# Astrocytic GABAergic Regulation in Alcohol Use and Major Depressive Disorders

**DOI:** 10.3390/cells13040318

**Published:** 2024-02-09

**Authors:** Dina N. Ali, Hossam M. Ali, Matthew R. Lopez, Shinwoo Kang, Doo-Sup Choi

**Affiliations:** 1Department of Molecular Pharmacology and Experimental Therapeutics, Rochester, MN 55905, USA; ali.dina@mayo.edu (D.N.A.); ali.hossam@mayo.edu (H.M.A.); lopez.matthew@mayo.edu (M.R.L.); kang.shinwoo@mayo.edu (S.K.); 2Neuroscience Program, Rochester, MN 55905, USA; 3Department of Psychiatry and Psychology, Mayo Clinic College of Medicine and Science, Rochester, MN 55905, USA

**Keywords:** GABA, astrocyte, alcohol use disorder, major depressive disorder, GABA transporter, GABA receptor, globus pallidus, amygdala

## Abstract

Gamma-aminobutyric acid (GABA) is the major inhibitory neurotransmitter in the central nervous system (CNS). Most GABAergic neurons synthesize GABA from glutamate and release it in the synaptic cleft in the CNS. However, astrocytes can also synthesize and release GABA, activating GABA receptors in the neighboring neurons in physiological and pathological conditions. As the primary homeostatic glial cells in the brain, astrocytes play a crucial role in regulating GABA homeostasis and synaptic neurotransmission. Accumulating evidence demonstrates that astrocytic GABA dysregulation is implicated in psychiatric disorders, including alcohol use disorder (AUD) and major depressive disorder (MDD), the most prevalent co-occurring psychiatric disorders. Several current medications and emerging pharmacological agents targeting GABA levels are in clinical trials for treating AUD and MDD. This review offers a concise summary of the role of astrocytic GABA regulation in AUD and MDD. We also provide an overview of the current understanding and areas of debate regarding the mechanisms by which astrocytes regulate GABA in the CNS and their potential significance in the molecular basis of AUD and MDD, paving the way toward future research directions and potential therapeutic target areas within this field.

## 1. Introduction

Alcohol use disorder (AUD) and major depressive disorder (MDD) are the most common co-occurring disorders substantially burdening worldwide well-being socially and financially [1,2,3]. Notably, AUD and MDD share many neurobiological underpinnings of symptoms [4,5,6,7,8,9,10,11]. Among several notable biological factors attributed to the pathophysiology of AUD and MDD, the dysregulation of γ-aminobutyric acid (GABA) levels and GABA receptor-mediated signaling are crucial to understanding specific phenotypes and, thereby, therapeutic targets [12,13,14]. As the primary inhibitory neurotransmitter in the brain, GABA is involved in multiple crucial brain functions such as learning and memory, motor coordination, emotion regulation, sensory processing, addictive behavior, and circadian rhythm [15,16,17,18,19,20,21]. In MDD patients, the biomarkers associated with GABA metabolism and signaling are altered in the cortical region [22]. While it is known that brain GABA deficits are attributed to the etiology of MDD, GABAergic involvement in the molecular mechanisms of AUD is less clear. Chronic alcohol administration decreases GABA-mediated responses in the cortex and nucleus accumbens, while alcohol increases the GABA released into the central nucleus of the amygdala, facilitating GABAergic transmission [23]. Furthermore, alteration in GABAergic signaling has been implicated in alcohol reinforcement effects and compulsive drinking in variable ways [24,25]. While the activation of GABA_B_ receptors is known to decrease the reinforcement actions of alcohol, acute blockade of GABA_A_ receptors can block the motivation for responding to alcohol [26,27,28,29]. GABA and GABA receptors have also been involved in the changes in the reward system associated with acute withdrawal [13,14].

Astrocyte and neuron interactions play a pivotal role in regulating and maintaining GABA levels in the brain [30]. Astrocytes, one of the glial cell types, account for at least half of the CNS cells [31], and have diverse functions, including but not limited to modulating synapse formation, function, and maturation, hence their vital role in synaptic plasticity [32,33,34] and maintaining and regulating CNS homeostasis through balancing the uptake and release of gliotransmitters and regulating ions (such as Ca^2+^, K^+^, H^+^) and water transport [35,36]. Gliotransmission, one of the critical functions of astrocytes in the brain, is basically the release of neurotransmitters from glial cells. The glial cells synthesize and/or store these neurotransmitters and regulate their release in response to pathophysiological stimuli, resulting in rapid responses (milliseconds to seconds) in target cells [37]. As shown in Figure 1, astrocytes can synthesize, take up, and release GABA through different mechanisms [38,39]. Astrocytes’ ability to regulate GABA transmission and the CNS excitatory/inhibitory balance underlies some of the molecular mechanisms of AUD and MDD. Evidence shows that astrocytic GABA receptors’ and transporters’ expression is altered in depression and different stages of AUD. Additionally, it has been documented that some of the preclinical and clinical therapeutics studied for AUD and MDD exhibit effects on astrocyte–neuron GABA signaling [40,41,42,43,44].

In this review, we first provide an overview of major advances in our understanding of astrocytes in regulating GABA homeostasis (Figure 1). We next discuss how diverse GABA signaling contributes to the molecular mechanisms underlying the pathogenesis of AUD and MDD. Lastly, we review some investigational drugs which target the GABA system and their potential as future clinical therapeutics in AUD and MDD.

## 2. Astrocytic Regulation of GABA in the CNS

GABA exerts its inhibitory effects by interacting with two distinct receptor types: GABA_A_ receptors (GABA_A_R) and GABA_B_ receptors (GABA_B_R). GABA_A_R is an ionotropic receptor and ligand-gated ion channel responsible for inhibitory synaptic transmission in the CNS [45,46,47]. GABA_B_R, a metabotropic receptor, operates at a slower pace through G-protein-coupled mediated signaling [48]. Immunohistochemical analysis of the adult human brain reveals that astrocytes express GABA_A_R and GABA_B_R at levels comparable to or even more significant than those observed in known GABAergic neurons. Additionally, cultured astrocytes derived from adult human brain tissue confirm the presence of GABA_A_R and GABA_B_R at both the mRNA and protein levels, establishing their dual GABAergic and GABAceptive characteristics [49]. In primary cell cultures and rodent slices, extracellular GABA can activate astrocytic GABAARs, increasing Cl^−^ concentrations in astrocytes. GABA also activates astrocytic GABA_B_Rs, mediating slow inhibitory signaling in the brain via the activation of Gi/o-type G-proteins, leading to the inhibition of adenylyl cyclase. The activation of astrocytic GABA_B_Rs increases intracellular Ca^2+^_,_ which triggers the release of Ca^2+^ from the intracellular pools. Ca^2+^ oscillations in astrocytes alter glutamate release and GABA transporter expression. Therefore, astrocytes can internalize GABA via GABA receptors and transporters, indicating their GABAceptive properties [50].

The regulation of GABA levels within the CNS is contingent upon the dynamic interaction between neurons and astrocytes (Figure 1). Both neurons and astrocytes are responsible for the synthesis, release, and reuptake of GABA within synapses. Each process responds to various pathophysiological stimuli to maintain GABA homeostasis. These processes are multifaceted, with distinct mechanisms operating in different cell types and regions of the brain [30,51,52,53,54,55]. Due to their close association with neurons and ability to engage in GABA synthesis, release, and reuptake, astrocytes are now considered pivotal contributors to the intricate task of regulating GABA homeostasis within the CNS [56]. Two glutamate decarboxylases, 1 and 2 (GAD1 and GAD2), produce GABA from glutamate. GAD1 is primarily expressed in neuronal cell bodies, while GAD2 is found in axon terminals [57]. Notably, GAD1, which plays a vital role in GABA synthesis, has also been observed in astrocytes [50,58]. Another crucial precursor of GABA in the CNS is putrescine, metabolized by monoamine oxidase B (MAO-B) and diamine oxidase (DAO). MAO-B is particularly prevalent in astrocytes, especially in cerebellar and striatal astrocytes [59]. Recent studies strongly suggest that GABA synthesis in astrocytes is a predominant factor in regulating GABA levels, suggesting that astrocytes are GABAergic cells [50].

The release of GABA from astrocytes occurs through three distinct mechanisms: calcium-dependent vesicular exocytosis, direct release into the extracellular space via GABA transporters (GATs) in reverse mode, or through GABA-permeable channels [51,60,61]. GATs function as secondary active electrogenic transporters and utilize sodium and chloride ion exchange with GABA uptake. Although the primary role of GATs is to remove excess GABA from the extracellular space, they were found to act in reverse mode, releasing GABA into the extracellular space in certain conditions, a function that remains controversial [56,62]. GATs are found in presynaptic neurons and astrocytes, with four known types (GAT1, GAT2, GAT3, and BGT1), among which GAT1 and GAT3 exhibit high affinity for GABA [61,62]. While GAT1 was thought to be primarily expressed in neurons, recent research shows that GAT1 is expressed in cortical and thalamic astrocytes as well. [63,64]. GAT3, on the other hand, is exclusively expressed in astrocytes predominantly localized at the astrocytic processes, modulating tonic inhibitory currents in postsynaptic cells. GAT3 activities influence various astrocytic functions encompassing the regulation of inhibitory synapse efficacy, excitatory neurotransmission, and astrocyte synaptic proximity, underscoring the role of GAT3 as a key glial GABA transporter [65,66]. Interestingly, the genetic ablation of GAT3 is lethal in mice, suggesting its critical role in GABA homeostasis during early embryogenesis and development [67].

Furthermore, astrocytic GABA release is facilitated by permeable membrane channels, such as BEST1, which is a calcium-dependent anion channel and is considered a vital mechanism of astrocytic GABA release [51]. Astrocytic GABA is known to interact with mainly surrounding neurons [50]. A recent study shows that astrocytes sense environmental change and release gliotransmitters, including GABA, to alter neuronal activities [68,69]. Astrocytic GABA can interact with synaptic and extrasynaptic receptors [60,68].

The GABA taken up by astrocytes via GABA transporters is either recycled into the GABA/glutamine cycle, which plays critical roles in GABA metabolism between neurons and astrocytes (Figure 2), or metabolized within mitochondria by the GABA transaminase (GABA-T) enzyme, which is present in both neurons and astrocytes [53,70]. Importantly, research has demonstrated that almost half of the released GABA is taken up and metabolized by astrocytes, underscoring their role in regulating GABA levels in the CNS [67]. Therefore, it is evident that astrocytes are considered essential cells in maintaining GABA homeostasis in the brain.

## 3. Astrocytes and GABA in Alcohol Use Disorder (AUD)

Alcohol use disorder (AUD) is a chronic, relapsing disease characterized by compulsive drug seeking despite negative consequences on an individual’s life [71]. Alcohol (or ethanol) exerts its toxicity through alterations in multiple neurotransmitter systems, including the GABA, serotonin, dopamine, glutamate, acetylcholine, and opioid systems [72]. The neurotransmitter imbalances result in malfunctioning brain circuits responsible for cognitive function, decision making, motivation, reward, affect, and the stress response [73]. Despite the dire health and psychosocial consequences, AUD continues to persist as one of the leading causes of death globally [74,75]. According to the most recent report from the National Institute of Health (NIH), the annual mortality rate due to alcohol-related causes exceeds 140,000 individuals, making it one of the leading four preventable factors contributing to fatalities in the United States [76]. AUD is typically associated with the development of tolerance, dependence, and the impairment of social and occupational functioning [77]. Recent research has suggested a significant role in the facilitation of GABAergic transmission in the addictive properties of alcohol [13,14,78]. It has been proposed that GABA plays a substantial role in the neuroadaptations linked to the progression from controlled alcohol consumption to excessive drinking characterized by dependence and relapse [79,80].

Evidence indicates that alcohol alters GABA-mediated responses in various brain regions, including the cortex and substantia nigra [81]. Recent studies have demonstrated that alterations in GABA transmission are particularly pronounced in regions implicated in the negative reinforcing aspects of alcohol, such as the ventral tegmental area (VTA), globus pallidus (GP), and the amygdala [82,83]. In experiments involving in vitro slice preparations from the central amygdala, acute alcohol exposure enhances GABA_A_ inhibitory postsynaptic currents. In rats chronically exposed to alcohol, an increase in evoked GABA release was observed [84]. Additionally, acute pharmacological inhibition of GABA_A_R function effectively diminishes the motivation for responding to alcohol [13].

Furthermore, the utilization of selective GABAB agonists has been shown to reduce alcohol self-administration in rats and mitigate the alcohol deprivation effect in alcohol-preferring rats [85]. Additional experiments have revealed that the combination of agonists and antagonists at the GABA_A_R benzodiazepine site leads to a noteworthy reduction in alcohol administration when administered into the amygdala [86,87,88].

Moreover, research employing pharmacological GABA agonists and antagonists has implicated the GABA system in both the physical and affective symptoms associated with alcohol withdrawal. GABA agonists were found to have the capacity to reduce CNS hyper-flexibility during alcohol withdrawal-induced seizures [89,90,91]. Consistently, GABA mimetics potentiate the sedative and motor effects of alcohol, an effect that was counteracted by GABA antagonists [24]. However, the underlying molecular mechanisms of the central effects of alcohol involving GABA-mediated signaling remain unclear. Notably, recent evidence has highlighted the role of astrocytes in modulating GABA transmission within the brain [30]. Chronic alcohol exposure alters the balance between inhibitory and excitatory neurotransmissions in various brain regions such as the cortex and the striatum [92]. Remarkably, a single astrocyte can modulate up to one million inhibitory and excitatory synapses. Consequently, any disruption to even a small subset of astrocytes can profoundly impact on the delicate balance between excitation and inhibition, ultimately, affecting brain function and behavior [93]. However, additional research may elucidate the precise role of astrocytes in ethanol-induced GABAergic neurotransmission.

An essential aspect of AUD is the disruption of the balance between goal-directed and habitual reward-seeking behaviors. Alcohol is known to exert effects on several signaling systems in the cortico-striatal circuits that may collectively contribute to the impairment of behavioral flexibility and motivate the transition from goal-directed to habitual alcohol drinking [94]. Changes in GABA release, uptake, and GABA receptor signaling across chronic alcohol exposure are critically involved in the acquisition of both goal-directed and habitual behaviors [95]. Studies indicate that chronic alcohol exposure may reduce the expression of GABA_A_R in the dorsal striatum, a region crucial for forming and expressing stimulus–response habits [96,97]. However, the precise direction in which alterations in GABA release and signaling drive behavior is unknown.

Given the pivotal role of astrocytes in regulating GABAergic transmission, recent studies have investigated the specific role of astrocytic GABAergic signaling in cognition and behavior. Recently, researchers documented that genetic ablation of GABA_B_R in the medial prefrontal cortex astrocytes altered the low gamma oscillations and firing properties of cortical neurons, affecting goal-directed behaviors [98].

Within the striatopallidal circuits, the dorsomedial striatum (DMS) and dorsolateral striatum (DLS) are the primary neural regions responsible for regulating goal-directed and habitual behaviors, respectively [99,100,101]. Nevertheless, the external globus pallidus (GPe), an area known to contain a substantial population of astrocytes, assumes a pivotal role in facilitating and coordinating the neurotransmission between the DMS and DLS, making it an integrative center for modulating the flexibility of reward-related behaviors (Figure 3) [102,103]. Notably, extrasynaptic neurotransmitters can trigger astrocyte Ca^2+^ signaling, and reciprocally, astrocyte Ca^2+^ signals modulate the function of the neural circuits through various gliotransmitters. Our recent research demonstrated that chemogenetic activation of astrocytes in the DMS differentially regulated striatal medium spiny neuron (MSN) activities and induced a shift from habitual to goal-directed reward-seeking behavior [101,104]. Despite these findings, it has not yet been fully elucidated how astrocytic modulation in the basal ganglia circuit may govern neuronal activities associated with goal-directed and habitual reward-seeking behavior in AUD. While the general role of astrocytes in regulating behavior, excitatory/inhibitory balance, and neuroplasticity in different brain regions is widely recognized, the specific involvement of astrocytic GABA signaling in the context of AUD remains unclear.

Our recent study has revealed that GPe astrocyte activity is suppressed during habitual learning in mice. Notably, chemogenetic activation of astrocytes has been shown to reduce habitual behaviors while concurrently enhancing goal-directed reward-seeking behaviors in operant conditioning experiments. Additionally, we found that activation of astrocytes reduced the overall activity of GPe neurons, which facilitated the transition from habitual to goal-directed alcohol-seeking behavior as well. Intriguingly, we observed an increase in GAT3 mRNA levels during habit formation, and the selective inhibition of GAT3 reversed the impact of astrocytes on the transition from habitual to goal-directed alcohol seeking. Our finding indicates that the upregulation of GAT3 in the GPe may deactivate astrocytes, diminishing their inhibitory influence on GPe neurons. These findings underscore the potential essential role of astrocytic GAT3 in regulating reward-related behavioral flexibility within the GPe [105]. Recent evidence supports the idea that GAT3 can govern astrocytic activity [106]. Nevertheless, the underlying mechanisms of GAT3 signaling, potentially involving other neurotransmitters, remain an area yet to be thoroughly explored. Moreover, given the diverse neuronal populations within the GPe, such as parvalbumin-expressing neurons, arkypallidal feedback, and prototypic feedforward neurons (Figure 3), further investigations on the specific changes related to GAT3 activity within distinct GPe cell types are warranted.

GABA transporters have recently garnered increasing attention within the field of addiction research, with a particular focus on GAT3, known for its high expression in astrocytes [66,107]. A recent study has uncovered a decrease in GAT3 mRNA levels in the amygdala, a forebrain structure recognized as a central hub for GABAergic influences on alcohol reinforcement, in alcohol-preferring mice. A reduction in several GABAAR subunits accompanied this decrease. The researchers proposed that this latter observation could indicate heightened GABAergic activity resulting from a decrease in the extracellular clearance of GABA. To further substantiate these findings, the researchers employed a viral GAT3 (*Slc6a11*) knockdown strategy in a mice group that initially preferred saccharin over alcohol. Their results revealed that, following the full expression of the injected virus in the amygdala, mice with GAT3 knockdown exhibited a shift in behavior from saccharin preference to alcohol preference [78].

The prolonged administration of alcohol, sufficient to induce dependence and escalate alcohol consumption, which is linked to increased GABA release within the amygdala, is accompanied by enhanced sensitivity to GABA agonists [108,109]. A study demonstrated that the microinjection of the GABAA agonist muscimol into the central nucleus of the amygdala (CeA) of alcohol-dependent rats reduced alcohol self-administration. However, this effect was not observed in alcohol nondependent rats [23,110]. Furthermore, a recent study revealed that rats with diminished GAT3 expression in the amygdala exhibited a propensity for persistent alcohol seeking, even when the alcohol was mixed with quinine. These findings suggest that extracellular GABA homeostasis in the amygdala plays a vital role in vulnerability to compulsive alcohol seeking. On the other hand, baclofen, a GABA_B_R agonist, lowers extracellular GABA levels in the amygdala and reduces alcohol consumption in mice and humans [111]. Interestingly, baclofen’s efficacy in diminishing the susceptibility to compulsive drinking and GAT3 expression in the amygdala are inversely correlated. This study showed a positive correlation between GAT3 mRNA levels in the amygdala and increased resistance to quinine in baclofen-treated but not vehicle-treated rats. This implies that baclofen’s effects may be mediated by normalizing impaired GABA clearance resulting from low GAT3 expression in the amygdala. However, it remains unknown whether solely restoring GABA homeostasis in the amygdala is sufficient to reverse compulsive alcohol seeking. Furthermore, additional research may be necessary to elucidate the psychological consequences of manipulating the GABAergic system within the amygdala and whether GAT3 expression is associated with baclofen’s known side effects. Collectively, these findings highlight the pivotal role played by astrocytic GAT3 in the molecular mechanisms of alcohol-seeking behavior and its potential as a therapeutic target for AUD.

## 4. Astrocytes and GABA in Major Depressive Disorder (MDD)

In 2021, the NIH’s estimated prevalence of MDD showed that around 15 million adults experienced at least one major depressive episode with severe impairment in the past year. MDD was reported as the most prevalent mental disorder [112] and the chief risk factor for suicide [113,114]. While the overarching cause of depression varies from one individual to another, one theory proposed that a combination of environmental factors, genetic susceptibility, and chronic stress might interact to disrupt neurotrophin signaling, resulting in impaired neurogenesis in the dentate gyrus and atrophy of distal dendrites, contributing to the genesis of depression [115]. Another popular theory regarding the biochemical bases of depression comes from the monoamine hypothesis, which suggests that an improper balance of the monoamine neurotransmitters, such as serotonin (5-HT), dopamine (DA), and norepinephrine (NE), is the primary cause of subsequent biological and psychological symptoms observed in individuals with MDD, which lead to the development of serotonin-specific reuptake inhibitors (SSRIs) as first-line management for MDD [116,117,118]. Despite the revolutionary effect of monoamine-derived antidepressants in the management of depression and their tremendous positive impact on patient outcomes, this hypothesis left many unresolved questions about both the cause and the treatment of MDD. Notably, only 30% of patients with MDD experience full remission with adequate treatment, and up to 46% of patients do not respond to first-line treatment, developing treatment-resistant depression (TRD). Recently, MDD was hypothesized to be a synaptic disorder rather than a simple imbalance of an individual neurotransmitter. The transition from the monoamine hypothesis to the neuroplasticity hypothesis initially focused on the dysregulation of excitatory glutamatergic synaptic transmission. Depression is associated with glutamate dysregulation in the prefrontal cortex, the amygdala, and the hippocampus [119]. As accumulating evidence suggests that MDD is associated with excitation/inhibition imbalance, in the past decade, numerous studies have emphasized the GABAergic mechanisms underlying MDD. In depression and chronic stress patients, a deficit of inhibitory synaptic transmission onto principal glutamatergic neurons in the prefrontal cortex was documented [12,22,120]. Both clinical and preclinical data support the association of MDD with direct defects in GABAergic neurotransmission. Laboratory work, including examinations of blood plasma, cerebral spinal fluid, and resected cortical tissue of patients with MDD, has consistently shown decreased GABA levels. More recent proton magnetic resonance spectroscopy studies have confirmed reduced GABA levels in several cortical regions [12,117,121]. Similarly, stress, a risk factor for depression, has been demonstrated to decrease prefrontal cortex GABA levels in animal models [22,122]. Furthermore, in rodent models of depression, both rats and mice have exhibited decreased GABA_A_R. Human MDD studies have also revealed decreased gene expression of GABAergic neuronal subtypes in cortical tissue, including somatostatin-, parvalbumin-, neuropeptide Y-, and calretinin-expressing neurons [12,22].

Given the prevailing hypothesis that the hypo-GABAergic system is implicated in MDD molecular mechanisms, both chemical (SSRI) and non-invasive stimulation such as transcranial magnetic stimulation (TMS) normalize the GABA levels [123,124]. More recent research examined the effects of the psilocybin analog 4-OH-DiPT, which reduced MDD-like symptoms in mice in a dose-dependent manner. 4-OH-DiPT ultimately increases GABAergic inhibition via binding to 5-HT2A receptors on the basolateral amygdala (BLA) GABAergic neurons [125].

The involvement of GABA dysfunction in the underlying pathology of MDD is a complicated, multifaceted theory. Earlier studies have focused on studying the phasic GABA released by inhibitory neurons. However, recently, it has become evident that tonic GABA, an additional form of neuronal inhibition, is also important in regulating the excitation/inhibition balance [40]. A more recent approach focused on the contribution of GABA tone to the interplay between the imbalance of excitatory/inhibitory transmission and a decrease in synaptic plasticity in the brain, hypothesized as a core mechanism underlying the pathology of MDD with potential therapeutic implications. The study found increased GABA tonic inhibition in the prefrontal cortex of the Flinders sensitive line (FSL) rat models of depression. Moreover, FSL rats showed reactive astrocytes and impaired plasticity in the cortex and hippocampus. Interestingly, the reactive astrocytes released more GABA, contributing to the increased tonic inhibition by activating extrasynaptic GABAAR. Furthermore, increased MAO-B enzyme activity in the same mice was observed, suggesting that astrocytes tend to synthesize more GABA through the MAO-B pathway in such pathological conditions. Using selegiline, a selective MAO-B-irreversible inhibitor, the astrocytic content of GABA was reduced, resulting in decreased GABA tonic inhibition. Additionally, selegiline restored LTP without affecting the synaptic inhibitory currents [44]. Given that selegiline is an antidepressant and is used clinically as a transdermal treatment through the activating monoamine system [126], these findings suggest that decreasing astrocytic GABA release in the prefrontal cortex is an alternative mechanism underlying the antidepressant effect of selegiline through restoring the excitation/inhibition imbalance and synaptic plasticity. However, inhibiting MAO-B could potentially alter the signaling of other neurotransmitters such as dopamine. Further research is necessary to elucidate the possible involvement of transmission signals other than astrocytic GABA in the alteration of synaptic plasticity implicated in the pathogenesis of MDD.

Previously, animal models for depression have reported reduced expression of GAT3. Zink and colleagues discovered that rats with congenital helpless behavior (cH), a genetic rat model for human depression, exhibited a significant decrease in GAT3 expression compared to non-helpless littermates [127]. Such downregulation might be attributed to the decreased GABA levels in mice with depression-like behaviors. Notably, they did not find downregulation of GAT1 or GAD67, indicating that GAT3 is critical for regulating GABA levels and supporting the hypothesis of impaired glial functions in depression. More recently, in mice exposed to chronic unpredictable mild stress, gene and protein expression of GAD1, VGAT, and GAT3 were reduced in the nucleus accumbens, an area targeted in the comorbidity of depression and addiction [128,129]. Another recent experiment suggested a strong correlation between GAT1/GAT3 and Parkinson’s disease (PD)-related depression. Additionally, selective block of GAT1 in the lateral habenula of PD rodent models increases the extracellular levels of GABA and produces more antidepressant responses in the PD mice group than in the wild-type group. However, blocking GAT3 produced antidepressant responses only in the PD mice group [130]. It is possible that the role of GABA transporters in the molecular basis of MDD may vary across individuals and specific subtypes of depression. Considering the minimal evidence in this matter, further research aiming to unravel the molecular and cellular mechanisms underlying the involvement of GABA homeostasis and astrocytic GABA transporters in MDD is warranted.

## 5. Potential Therapeutic Targets in AUD and MDD

### 5.1. AUD

As summarized in Table 1, preclinical and clinical research involving medications that target the GABA system in the context of AUD is an active area of study. Some preclinical studies have investigated the potential of GABA receptor agonists to reduce alcohol consumption. These medications enhance the action of GABA, which can help reduce the reinforcing effects of alcohol. Baclofen, a GABABR agonist, has shown promise in reducing alcohol intake in animal models and clinical trials (Table 1). According to the current evidence, baclofen is more effective than a placebo at decreasing days of heavy drinking and increasing days of abstinence in people with alcohol dependency [131]. With medium effect sizes, the effect was most prominent in subjects receiving 90 mg/day of baclofen [132]. However, baclofen meta-analyses have produced contradictory findings; therefore, the current findings cannot be used to estimate the overall impact size of baclofen in AUD [133]. The current investigation broadens the corpus of research, suggesting that baclofen may be helpful in the management of patients with alcohol use problems [134]. There are still unanswered concerns about the effect sizes for various outcomes in large populations with AUD and the existence of response modifiers that might aid in the prescription of guidelines. At 30 mg of baclofen per day, a noteworthy impact was found on women. Also, it has been suggested that baclofen could be beneficial for reducing anxiety with AUD, an effect that remains controversial [131,135,136,137]. A daily dose of 90 mg demonstrated superior effectiveness overall, but it was associated with decreased tolerance in women. Gender appears to influence treatment response, with men experiencing positive effects from a daily dose of 90 mg of baclofen but not from 30 mg/day. In contrast, women benefited from a 30 mg/day dose of baclofen, saw some improvement with 90 mg/day, but also experienced increased intolerance at the higher dose [138].

Gabapentin is an analog of GABA that binds to the α2δ1 subunit of voltage-gated calcium channels, reducing excitatory postsynaptic currents (Table 1). Gabapentin has shown potential benefits in managing mild alcohol withdrawal. It has been found to improve residual craving and sleep measures, which are significant factors in preventing relapse [139]. The medication also shows promise in improving mood and anxiety, indicating its therapeutic effect. However, its effectiveness in moderate-to-severe alcohol withdrawal is yet to be established. Seizures have been reported during withdrawal despite gabapentin treatment, although it is unclear whether this is due to an insufficient dose, patient susceptibility, or lack of efficacy [140]. While evidence shows that gabapentin is not an agonist at GABA_A/B_ receptors, it may increase whole-brain GABA through an undefined mechanism [141,142].

**Table 1 cells-13-00318-t001:** Potential therapeutic medications for alcohol use disorder. The table includes some pharmacological agents targeting the GABA system currently under preclinical experiments or clinical trials for their potential use in treating AUD.

Medication	Model	Target	Clinical Implications	References
Baclofen	Clinical Studies	GABA_B_ receptor agonist	Reduces alcohol consumption and preference and decreases withdrawal symptoms in alcohol-dependent individuals; also supports the maintenance of abstinence from alcohol.	[131,132,134,135,138]
Benzodiazepines *	Rodents and Clinical Studies	GABA_A_ receptor modulators	Can reduce the symptoms of alcohol withdrawal syndrome and reduce alcohol intake and alcohol seeking.	[143,144]
Allopregnanolone	Rodents and Clinical Studies	Neuroactive steroid	Serves as a safeguarding element in healthy control individuals, reducing the risk of developing AUD.	[145]
Vigabatrin	Rodents	GABA-transaminase inhibitor	Diminishes ethanol reinforcement and amplifies the discriminative stimulus effect of ethanol, leading to a significant decrease in ethanol consumption.	[146]
KK-92A	Rodents	Positive allosteric modulator of the GABA_B_ receptor	Suppresses operant alcohol self-administration and reinstatement of alcohol seeking.	[147,148]
Semaglutide	Rodents and Clinical Studies	GLP-1 analogue increases GABA transmission in pyramidal neurons in layer 5 of the infralimbic cortex (ILC) and elevates dopamine levels in nucleus accumbens.	Decreases alcohol intake across different drinking models as it reduces alcohol intake and prevents relapse-like drinking.	[149]
Gabapentin	Clinical Studies and Rodents	Structural analog of GABA that binds to the alpha-2-delta type 1 subunit of voltage-gated calcium channels, reducing excitatory postsynaptic currents.	Most effective when implemented following the commencement of abstinence to maintain it, with its optimal performance likely observed in individuals with a track record of more intense alcohol withdrawal symptoms.	[139,140,150,151,152,153]
Muscimol	Rodents	GABA_A_ receptor agonist	Intra-amygdala muscimol had a significant inhibitory effect on alcohol-seeking behavior in alcohol-dependent rats but had no impact on nondependent controls. In addition, it ameliorated the sleep–wake disruptions in alcohol-withdrawn rats by reducing the percentage of active wakefulness and increasing the percentage of REM sleep.	[110,154]
SR-95531	Rodents	GABA_A_ receptor antagonist	Decreased oral ethanol-seeking response in rats.	[87]
Tiagabine	Clinical Studies	Selective inhibitor of GABA reuptake by transporter subtype (GAT-1)	May reduce alcohol consumption and decrease alcohol dependence.	[155]

* Benzodiazepines includes diazepam, chlordiazepoxide, and lorazepam.

Allopregnanolone is a neuroactive steroid that can modulate GABA_A_R as allosteric modulators (Table 1). Preclinical studies have investigated the potential of synthetic neurosteroids to influence alcohol-related behaviors. Allopregnanolone is a neurosteroid that has shown promise in reducing alcohol consumption and withdrawal symptoms in animal models and could serve as a safeguarding element in healthy control individuals, reducing the risk of developing AUD [145].

It is important to note that while preclinical research can provide valuable insights into potential treatments for AUD, translating these findings to clinical medications safe for humans is a complex process involving rigorous testing for safety and efficacy. Many medications that show promise in animal models may not prove effective in human trials or have significant side effects. Therefore, further research, including clinical trials, is necessary to determine the viability of these medications as treatments for AUD.

### 5.2. MDD

GABAergic medications have emerged as a promising yet complex avenue of research in MDD. While the GABAergic system’s inhibitory function in the brain and its dysregulation in MDD have raised interest, translating preclinical findings into clinical practice remains challenging. As summarized in Table 2, clinical trials evaluating GABAergic medications have produced mixed results with varying efficacy and tolerability. Additionally, concerns about the risk of dependence, moderate abuse potential, and the danger of withdrawal symptoms, especially with benzodiazepines, underscore the need for careful consideration of their use [156]. Benzodiazepines bind to the GABA_A_R at allosteric sites and enhance the GABA_A_R current by increasing chloride conductance. Benzodiazepines binding to the synaptic GABA_A_R induce a conformational change for which GABA has a higher affinity, thereby increasing the frequency of chloride channel opening [157,158,159]. Benzodiazepines are sometimes used to treat specific symptoms associated with depression, such as anxiety and insomnia. However, a meta-analysis study showed that treatment with the benzodiazepine alprazolam led to a higher percentage of individuals with MDD achieving response on the 17-item Hamilton Depression Rating Scale (HAM-D) or the Clinical Global Impression—Improvement (CGI-I) scale, compared with placebo. Other benzodiazepines, such as diazepam, did not exhibit a clear antidepressant effect [160].

In addition, prolonged use of benzodiazepines beyond 2–4 weeks is not recommended due to potential decrease in GABAergic and monoaminergic function, interference with neurogenesis, and cognitive and psychomotor impairment [161,162]. These concerns, along with the elevated risk of dependence and suicide attempts, might further restrict the potential use of benzodiazepines in the treatment of depression [163].

Tiagabine is a GABA reuptake inhibitor that has been studied for its potential as an augmentation strategy in treatment-resistant depression (Table 2). Also, it has been hypothesized that it enhances GABA tone and is currently being assessed for its efficacy and safety in the treatment of depression comorbid with anxiety [164].

Brexanolone (SAGE-547) and zuranolone (SAGE-217) are investigational drugs that target GABA_A_Rs and have shown promise in clinical trials for postpartum depression [165] (Table 2). Brexanolone and zuranolone are synthetic derivatives of the neuroactive steroid allopregnanolone, working as positive allosteric modulators of synaptic and extrasynaptic GABA_A_R with the capacity to augment GABAergic neurotransmission [166]. They were initially conceived to address postpartum depression (PPD) and are presently under scrutiny for their potential in the context of MDD. Clinical investigations into their efficacy are still in progress [167]. Additional backing for this assertion comes from research indicating that the administration of allopregnanolone prevented or restored depressive or anxiety-related behaviors in a rodent model of social isolation [168].

**Table 2 cells-13-00318-t002:** Potential therapeutic medications for major depressive disorder. The includes some of the GABA-targeted pharmacological agents that are currently studied clinically and preclinically for their potential use in treating MDD.

Medication	Model	Target	Clinical Implications	References
Tiagabine	Clinical Studies and Rodents	Selective inhibitor of GABA reuptake by transporter subtype (GAT1)	Demonstrated antidepressant-like properties in animal models.	[164,169]
* Brexanolone and zuranolone	Clinical Studies	Neuroactive steroids and positive allosteric modulators (PAM) of GABA_A_ receptors	Zuranolone reduced depressive symptoms after 2 weeks of use. While brexanolone has been studied and approved for postpartum depression.	[165,166,167,170]
PRAX-114	Clinical Studies and Rodents	Extrasynaptic GABA_A_ receptor positive allosteric modulator	Attained antidepressant-like effects that varied based on the dosage.	[171,172]

* Brexanolone is SAGE-547; Zuranolone is SAGE-217.

While ketamine’s rapid antidepressant effect is primarily attributed to increased glutamate connectivity and signaling, evidence suggests that it induces a transient increase in GABA signaling shortly after administration in rodents [173]. Imaging studies report that ketamine normalizes the hyperactivity of the default mood network (DMN) and altered connectivity with the insula in depressed patients up to 2 days after infusion, potentially due to enhanced GABA function [174]. Studies in healthy controls support the hypothesis that ketamine enhances GABA inhibition, as evidenced by reduced DMN connectivity and decreased reactivity of the amygdala–hippocampal circuitry in response to emotional stimuli in ketamine-treated patients [174,175,176]. Moreover, during ketamine infusion, GABA responses in the medical prefrontal cortex were measured as ratios relative to unsuppressed voxel tissue water (W). It was found that ketamine increased the GABA/water ratio during infusion in depressed patients [177]. Both ketamine and its enantiomer, (R)-ketamine, have exhibited swift-acting antidepressant properties and were the subject of active investigation for their potential application in the treatment of MDD [178,179,180].

PRAX-114 is an oral formulation of a primarily extrasynaptic GABA_A_R-positive allosteric modulator (PAM), which is under investigation for the treatment of MDD (Table 2). A phase 2 study conducted in Australia yielded interim results that demonstrated improvements in depression severity [172]. Furthermore, a phase 2/3 trial, randomized, double-blind, and placebo-controlled, involving 216 participants, assessed the safety and efficacy of a 28-day monotherapy treatment course with PRAX-114 for severe MDD [172,181]. Unfortunately, this trial has not been continued as the sponsor has no further intentions to pursue the development of PRAX-114 for psychiatric disorders. Regardless, the efforts to develop medications for MDD by modulating GABAergic receptors and GABA levels will be continued.

Further research is required to delineate the specific subpopulations of MDD patients who might benefit from GABAergic medications and to explore their role in combination with other therapeutic strategies. While the potential of GABAergic medications is tantalizing, their integration into the treatment arsenal for MDD demands a critical assessment of both their advantages and limitations.

## 6. Discussion

This review provides insight into how neurons and astrocytes regulate GABA levels to exert multiple physiological and pathological neurotransmission modulations in AUD and MDD. While neuronal GABAergic signaling constitutes the primary synaptic activities in various local circuits involving AUD and MDD, we emphasize the importance of neuron–astrocyte interaction and astrocytic GABA release and transport in the basal ganglia circuits. In particular, we highlight the role of regulating astrocytic GAT3 in rodents and humans in AUD [102]. Our recent study also demonstrated that GAT3 is critical in regulating GABAergic signaling in reward- and alcohol-seeking behaviors [105]. Not surprisingly, GABA levels and GABA receptors are responsible for depression-related phenotypes and treatment targets, as we discussed with regard to the common pathophysiology of AUD and MDD. Among several brain regions and circuits, we show a role of astrocytic GABAergic regulation in the GPe, which is known to integrate GABAergic inputs from the dorsal striatum, including caudate-like dorsomedial striatum (DMS) and putamen-like dorsolateral striatum (DLS), which also regulate goal-directed and habitual reward-seeking behaviors, respectively [105]. The main output of the GPe is the subthalamic nucleus (STN) and the internal globus pallidus (GPi); the GPe is known as a hub of action selection and behavioral flexibility [100,182]. Notably, the GPe contains abundant astrocytes, which may orchestrate different GABAergic neurons. Our recent studies show that the GPe astrocyte–neuron interactions are critical for transitioning goal-directed and habitual reward-seeking behaviors [105]. We summarized the medications currently used and under development for AUD and MDD targeting the GABAergic system. Having considered that only three FDA drugs are available in AUD treatment while more than 20 drugs are available for MDD, the need for more potent and efficacious drugs for AUD is demanding [183]. Two recent advances in drug development are notable. One direction is to use biased allosteric modulators in targeting GABA_A_ and GABA_B_ receptors to specify the downstream signaling and elicit the desired pharmacological efficacy, as listed in Table 1 and Table 2. Although there are some challenges in the clinical trials and FDA approval of positive and negative allosteric modulators for AUD and MDD, future advancements and refinements of drug design and clinical trials will shed light on promising drugs with minimal side effects. On the other hand, we also need to identify biomarkers and clinical determinants associated with existing and emerging medications in AUD and MDD. Although we have not focused on this topic in our review, precision therapy with improved efficacy and reduced side effects will eventually benefit diverse and heterogeneous AUD and MDD patient populations.

## 7. Conclusions

This review updates the role of astrocytes in regulating GABA homeostasis through the intricate mechanisms of GABA synthesis, release, and uptake. We summarize the GABAergic and GABAceptive properties of astrocytes. We provide recent findings regarding astrocytes’ ability to regulate synaptic plasticity and behavioral flexibility, focusing on AUD and MDD. This review highlights a concise summary of the current, albeit limited, evidence demonstrating how the astrocytic expression of GABA receptors and transporters may provide insights further explaining the underlying mechanisms of AUD and MDD, ultimately paving the way toward future research directions and potential therapeutic medications. Additionally, this review outlines some of the ongoing clinical and preclinical investigations into potential GABA-targeted medications in AUD and MDD, highlighting some of their limitations. Future research will elucidate the precise and comprehensive mechanisms of astrocytic GABAergic regulation underlying the pathophysiology of MDD and AUD.

## Figures and Tables

**Figure 1 cells-13-00318-f001:**
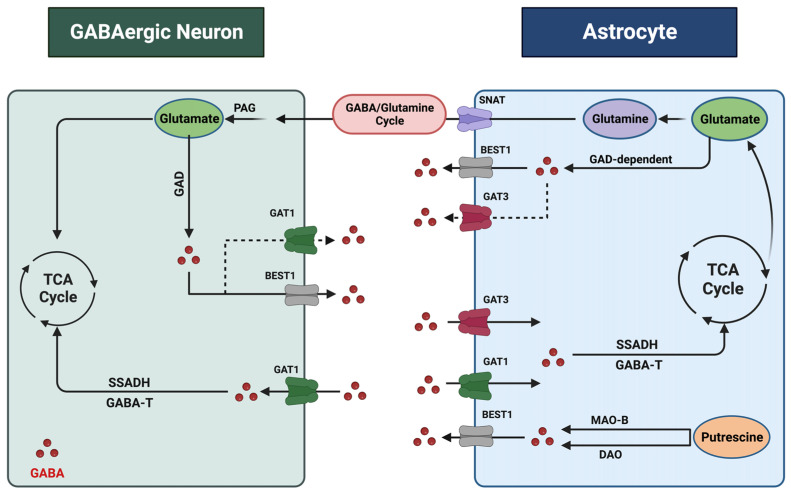
Astrocyte–GABAergic neuron interaction. GABA undergoes extensive recycling between neurons and astrocytes, a process coordinated and regulated by cell-specific transporters and enzymes. Within astrocytes, GABA and glutamate are metabolized, supporting the synthesis of the non-neuroactive amino acid glutamine. Astrocyte-derived glutamine is taken up by neurons and converted to glutamate by phosphate-activated glutaminase (PAG) to replenish the neurotransmitter pool. Sodium-coupled neutral amino acid transporters (SNATs) facilitate glutamate transfer. Astrocytes synthesize GABA from glutamate through a glutamate decarboxylase (GAD)-dependent pathway. Putrescine serves as another precursor for GABA, and monoamine oxidase B (MAO-B) and diamine oxidase (DAO) are the key enzymes in this pathway. Additionally, the BEST1 (bestrophin 1) channel mediates GABA release. GABA transporters recycle excess GABA from the synapse through uptake into neurons and astrocytes. Astrocytes express GAT1 and GAT3, while GABAergic neurons express GAT1. Following uptake, intracellular GABA is metabolized by the enzymes GABA transaminase (GABA-T) and through the intermediate succinate semi-aldehyde by the enzyme succinate semi-aldehyde dehydrogenase (SSADH). This figure was created with BioRender.com (accessed on 17 November 2023).

**Figure 2 cells-13-00318-f002:**
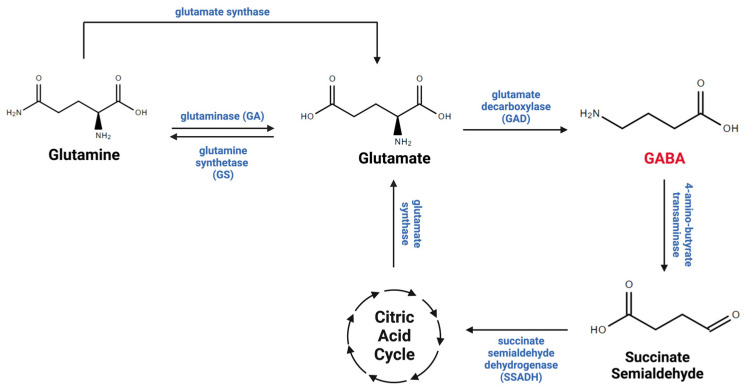
GABA metabolism. GABA is metabolized through the GABA/glutamate/glutamine cycle, which connects neurotransmitter homeostasis and cellular energy metabolism. Glutamate is the principal biological precursor for GABA through the enzyme glutamate decarboxylase (GAD). GABA is converted to succinic semi-aldehyde, which is metabolized into the tricarboxylic acid (TCA) cycle through the enzyme succinic semi-aldehyde dehydrogenase (SSDAH). GABA, glutamate, and glutamine undergo oxidation within the TCA cycle, actively contributing to energy production in neurons and astrocytes. This figure was created with BioRender.com (accessed on 17 November 2023).

**Figure 3 cells-13-00318-f003:**
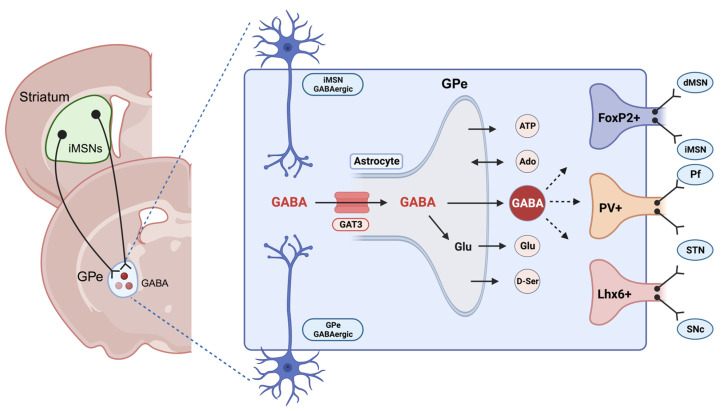
GABA transport in the globus pallidus externus (GPe) neurons. The “GPe” accommodates diverse neuronal populations, such as parvalbumin+ neurons and arkypallidal neurons expressing FoxP2+ and LIM homeobox-positive (Lhx6+) neurons. GABAergic neurons in the GPe receive inputs from the striatal indirect pathway medium spiny neurons (iMSNs). The GPe comprises numerous astrocytes capable of releasing neurotransmitters such as ATP, adenosine, glutamate, D-serine, and GABA [82,83]. GABA transporter 3 (GAT3) is an astrocyte-specific transporter responsible for either releasing into the synapse or metabolizing into the GABA/glutamine cycle. This figure was created with BioRender.com (accessed on 17 November 2023).

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
