# Peer review of "Astrocytic GABAergic Regulation in Alcohol Use and Major Depressive Disorders"

_cells, 2024, doi:10.3390/cells13040318_

Round 1

Reviewer 1 Report

Comments and Suggestions for Authors

In this manuscript, the authors conducted extensive literature review on astrocytic GABAergic regulation in alcohol use disorders (AUD) and major depressive disorders (MDD) and provided a summary of the role and mechanisms of astrocytic GABA regulation in AUD and MDD. Overall, it is an interesting paper, well organized and nicely written. The whole paper needs a quick copy edit. There are a few typos, for example, the word “in" is missing from heading 4 on page 7 and the discussion section should be numbered 6 on page 11.

Author Response

Response: Thank you for the reviewer’s comments. We corrected all the typos.

Reviewer 2 Report

Comments and Suggestions for Authors

The article describes a detailed understanding of astrocytes in regulating GABA homeostasis. The review covers elaborative literature and recent findings on GABA synthesis, metabolism, and GABA receptor functions in astrocytes and neurons of the central nervous system. More importantly, the review highlights GABA, its receptor’s role in AUD and MDD, and relevant preclinical and clinical research.  The review is well-written, and the current version is acceptable for publication.

Author Response

Response: Thank you for the reviewer’s note.

Reviewer 3 Report

Comments and Suggestions for Authors

The review article from Ali et al., claims the importance of astrocytic function in GABA regulation in particular pathomechanisms of alcohol use disorder and major depressive disorder. Notwithstanding the obvious importance of astrocytes in neuronal function, and the interesting description of the role of astrocytes in GABA synthesis, reuptake and release, the development of the review is disparate and not always convincing. 

Specific points:

1) Introduction Section is far too concise, and the association of general epidemiologic aspects of AUD, MDD and their mutual comorbidity with another paragraph generally describing astrocyte contribution to GABA metabolism is falling short, seeming unjustified and weird as to this reviewer, it is not sufficiently explained.

2) In Section 2 the authors insist on the dual role of astrocytes as both GABAergic and GABAceptive cells. This is an interesting role, but it needs to be deeply described. For instance: which is the fate of GABA released by astrocytes? Which is the difference between an inhibitory synapse and the release of GABA by astrocytes? Astrocyte-limited GAT3 expression represents at this point the major proof in support of astrocyte contribution to all the suggested processes thereafter described.

3) Section 3 seems the most interesting of the review, however it is very confusing. Striatal control of habitual behavior represents a complex domain. The majority of readers will find it simply impossible to understand the proposed implication of astrocytes and their GABAergic function without a paragraph systematically describing fundamental mechanistic aspects and relative advancement in this field.

The second description of AUD features after the introductory one is simplistic, taking into consideration the complexity, and variety of AUD neuroplastic correlates. The most relevant issue, to this reviewer, is the real contribution of astrocytes to GABAergic aspects of ethanol addiction. It is out of doubt that GABA plays a fundamental role in AUD but the notion that GAT3 manipulation modulates addictive behaviors is insufficient to this reviewer to unambiguously define the astrocytic role as “central”, even more important than neurons, reading through this work. 

4) Section 4 show another simplistic description of MDD, monoamine hypothesis of depression is just one among the many, for example, the glutamatergic or the morphoplastic hypotheses are far more known and probably relevant that the GABAergic hypothesis. 

5) The notions and current proof that Gabapentin may represent an interesting pharmacological therapy for AUD totally emancipates from the evidences of GABA implications in AUD. As also acknowledged in the article, the target of Gabapentin, as well as of Pregabalin and Topiramate, all compounds proposed to exert a positive effect on AUD, are the voltage gated calcium channels, mainly found on glutamatergic neurons. So the question of this referee is which can be the relevance to describe Gabapentin activity in the light of astrocytes and GABA participation to AUD.

6) The first line medication for MDD are antidepressant drugs. It is interesting to mention allosteric GABA receptor regulators and their potential contribution to MDD management, but the limits and frequent contraindication of GABAergic manipulation in depression should be clearly defined. Ketamine again, displays a fast antidepressant effect which depend on specific homeostatic regulation of the excitatory synapse, as perfectly described in the works of Lisa Monteggia. The link between ketamine, astrocytes and GABA is weak, also taking into consideration that this referee does not believe its activity is contributed by inhibition of NMDA receptors at the level of inhibitory neurons.

Throughout the paper english mistakes and inappropriate sentences are abundant. For instance: line 72-73 unclear sentence.

Line 248 title of section 4. Line 269 this is absolutely not the consensus. Line 276 “congenital helpless behavior”, what is it? Line 278 “Levels of GABA levels”. Line 422, to be rewritten.

All these considerations negatively affect my decision about to accept this work for publication.

Comments on the Quality of English Language

English is predominantly acceptable, asking nonetheless a serious editing process.

Author Response

Dear Reviewer 3

We thank you for your comments, corrections, and suggestions. 

Reviewer #3: The review article from Ali et al., claims the importance of astrocytic function in GABA regulation in particular pathomechanisms of alcohol use disorder and major depressive disorder. Notwithstanding the obvious importance of astrocytes in neuronal function, and the interesting description of the role of astrocytes in GABA synthesis, reuptake and release, the development of the review is disparate and not always convincing. 

Specific points:

  • Introduction Section is far too concise, and the association of general epidemiologic aspects of AUD, MDD and their mutual comorbidity with another paragraph generally describing astrocyte contribution to GABA metabolism is falling short, seeming unjustified and weird as to this reviewer, it is not sufficiently explained.

Response: In the revised manuscript, we added two additional paragraphs to guide readers on the scope of our review articles as follows.

[lines 37-48] “In MDD patients, the biomarkers associated with GABA metabolism and signaling are altered in the cortical region [1]. While it is known that brain GABA deficits are attributed to the etiology of MDD, GABAergic involvement in the molecular mechanisms of AUD is less clear. Chronic alcohol administration decreases GABA-mediated responses in the cortex and nucleus accumbens, while alcohol increases GABA release into the central nucleus of the amygdala, facilitating GABAergic transmission [2]. Also, alteration in GABAergic signaling has been implicated in alcohol reinforcement effects and compulsive drinking in variable ways [3, 4]. While activation of GABAB receptors is known to decrease the reinforcement actions of alcohol, acute blockade of GABAA receptors can block the motivation for responding to alcohol [5-8]. GABA and GABA receptors have been involved in the changes of reward system associated with acute withdrawal [9, 10].”

[lines 60-65] “Astrocytes’ ability to regulate GABA transmission and the CNS excitatory/inhibitory balance underlies some of the molecular mechanisms of AUD and MDD. Evidence shows that astrocytic GABA receptors and transporters expression is altered in depression and different stages of AUD. Additionally, it has been documented that some of the preclinical and clinical therapeutics studied for AUD and MDD exhibit effects on astrocyte-neuron GABA signaling [11-15].”

  • In Section 2 the authors insist on the dual role of astrocytes as both GABAergic and GABAceptive cells. This is an interesting role, but it needs to be deeply described. For instance: which is the fate of GABA released by astrocytes? Which is the difference between an inhibitory synapse and the release of GABA by astrocytes? Astrocyte-limited GAT3 expression represents at this point the major proof in support of astrocyte contribution to all the suggested processes thereafter described.

Response: We thank you for the reviewer’s comments. We clarified by adding the following sentence:

[lines 141-144] “ Astrocytic GABA is known to interact with mainly surrounding neurons [16]. A recent study shows that astrocytes sense environmental change and release gliotransmitters, including GABA, to alter neuronal activities [17]. Astrocytic GABA can interact with synaptic and extra-synaptic receptors [18].”

[lines 91-104] “Immunohistochemical analysis of the adult human brain reveals that astrocytes express the GABAAR and GABABR comparable to or even more significant than those observed in known GABAergic neurons. Additionally, cultured astrocytes derived from adult human brain tissue confirm the presence of GABAAR and GABABR at both mRNA and protein levels, establishing their dual GABAergic and GABAceptive characteristics [19]. In primary cell cultures and rodent slices, extracellular GABA can activate astrocytic GABAARs, increasing Cl- concentrations in astrocytes. GABA also activates astrocytic GABABRs, mediating slow inhibitory signaling in the brain through the activation of Gi/o type G-proteins, leading to the inhibition of adenylyl cyclase. The activation of astrocytic GABABRs increases intracellular Ca+2, which triggers the release of Ca+2 from the intracellular pools. Ca+2 oscillations in astrocytes alter glutamate release and GABA transporters expression. Therefore, astrocytes can internalize GABA via GABA receptors and transporters, indicating their GABAceptive properties [16]."

[lines 132-136] “GAT3, on the other hand, is exclusively expressed in astrocytes and predominantly localizes to astrocytic processes, which modulates tonic inhibitory currents in postsynaptic cells. GAT3 activities influence various astrocytic functions encompassing the regulation of inhibitory synapse efficacy, excitatory neurotransmission, and astrocyte synaptic proximity, underscoring the role of GAT3 as a key glial GABA transporter [30]

  • Section 3 seems the most interesting of the review, however it is very confusing. Striatal control of habitual behavior represents a complex domain. The majority of readers will find it simply impossible to understand the proposed implication of astrocytes and their GABAergic function without a paragraph systematically describing fundamental mechanistic aspects and relative advancement in this field. The second description of AUD features after the introductory one is simplistic, taking into consideration the complexity, and variety of AUD neuroplastic correlates. The most relevant issue, to this reviewer, is the real contribution of astrocytes to GABAergic aspects of ethanol addiction. It is out of doubt that GABA plays a fundamental role in AUD but the notion that GAT3 manipulation modulates addictive behaviors is insufficient to this reviewer to unambiguously define the astrocytic role as “central”, even more important than neurons, reading through this work. 

Response: To clarify the section, we revised the section as follows.

[lines 201-207] “Chronic alcohol exposure alters the balance between inhibitory and excitatory neurotransmission in various brain regions such as the cortex and the striatum. Remarkably, a single astrocyte can modulate up to one million inhibitory and excitatory synapses. Consequently, any disruption to even a small subset of astrocytes can profoundly impact the delicate balance between excitation and inhibition, ultimately affecting brain function and behavior [31]. However, additional research may elucidate the precise role of astrocytes in ethanol-induced GABAergic neurotransmission."

[lines 218-222] “Given the pivotal role of astrocytes in regulating GABAergic neurotransmission, recent research investigated the specific implications of astrocytic GABAergic signaling in cognition and behavior. Recently, researchers documented that genetic ablation of GABABR in the medial prefrontal cortex astrocytes altered low gamma oscillations and firing properties of cortical neurons, affecting goal-directed behaviors [32].”

[lines 237-247] “Notably, extrasynaptic neurotransmitters can trigger astrocyte Ca+2 signaling, and reciprocally, astrocyte Ca+2 signals modulate the function of the neural circuits through various gliotransmitters. Our recent research demonstrated that chemogenetic activation of astrocytes in DMS differentially regulated striatal medium spiny neurons (MSNs) activities and induced a shift from habitual to goal-directed reward-seeking behavior [33, 34]. Despite these findings, it is not yet fully elucidated how astrocytic modulation in the basal ganglia circuit may govern neuronal activities associated with goal-directed and habitual reward-seeking behavior in AUD. While the general role of astrocytes in regulating behavior, excitatory/inhibitory balance, and neuroplasticity in different brain regions is widely recognized, the specific involvement of astrocytes in the context of AUD remains unclear.”

  • Section 4 show another simplistic description of MDD, monoamine hypothesis of depression is just one among the many, for example, the glutamatergic or the morphoplastic hypotheses are far more known and probably relevant that the GABAergic hypothesis. 

Response: We thank the reviewer’s comment. However, as stated in the original paragraph, we are not emphasizing the monoamine hypothesis. We wanted to review focusing on GABAergic signaling in astrocytes. We added additional sentences.

[lines 306-331] “While the overarching cause of depression varies from one individual to another, one theory proposed that a combination of environmental, genetic susceptibility and chronic stress might interact to disrupt neurotrophin signaling, resulting in impaired neurogenesis in the dentate gyrus and atrophy of distal dendrites contributing to the genesis of depression [35]. Another popular theory regarding the biochemical bases of depression comes from the monoamine hypothesis, which suggests that an improper balance of the monoamine neurotransmitters, such as serotonin (5-HT), dopamine (DA), and norepinephrine (NE), is the primary cause of subsequent biological and psychological symptoms observed in individuals with MDD, which lead to the development of serotonin specific reuptake inhibitors (SSRIs) as first-line management for MDD [36-38]. Despite the revolutionary effect of monoamine-derived antidepressants in the management of depression and their tremendous positive impact on the patient's outcomes, this hypothesis left many unresolved questions about both the cause and the treatment of MDD. Notably, only 30% of patients with MDD experience full remission with adequate treatment, and up to 46% of patients do not respond to first-line treatment, developing treatment-resistant depression (TRD). Recently, MDD is a synaptic disorder rather than a simple imbalance of an individual neurotransmitter. The transition from the monoamine hypothesis to the neuroplasticity hypothesis initially focused on the dysregulation of excitatory glutamatergic synaptic transmission. Depression is also associated with glutamate dysregulation in the prefrontal cortex, the amygdala, and the hippocampus [39]. As accumulating evidence suggests that MDD is associated with excitation/inhibition imbalance, in the past decade, numerous studies emphasized GABAergic mechanisms underlying MDD. In depression and chronic stress patients, a deficit of inhibitory synaptic transmission onto principal glutamatergic neurons in the prefrontal cortex was documented [1, 40, 41]. Both clinical and preclinical data support the association of MDD with direct defects in GABAergic neurotransmission.”

  • The notions and current proof that Gabapentin may represent an interesting pharmacological therapy for AUD totally emancipates from the evidences of GABA implications in AUD. As also acknowledged in the article, the target of Gabapentin, as well as of Pregabalin and Topiramate, all compounds proposed to exert a positive effect on AUD, are the voltage gated calcium channels, mainly found on glutamatergic neurons. So the question of this referee is which can be the relevance to describe Gabapentin activity in the light of astrocytes and GABA participation to AUD.

Response: Agreeing with the reviewer about Gabapentin, we removed the detailed description of Gabapentin (see the highlighted main text lines 436 to 457) and revised the paragraph as follows.

[lines 418-427] “Gabapentin is an analog of GABA that binds to the a2d1 subunit of voltage-gated calcium channels, reducing excitatory postsynaptic currents (Table 1). Gabapentin has shown potential benefits in managing mild alcohol withdrawal. It has been found to improve residual craving and sleep measures, which are significant factors in preventing relapse [42]. The medication also shows promise in improving mood and anxiety, indicating its therapeutic effect. However, its effectiveness in moderate to severe alcohol withdrawal is yet to be established. Seizures have been reported during withdrawal despite gabapentin treatment, although it is unclear whether this is due to an insufficient dose, patient susceptibility, or lack of efficacy [43]. While not an agonist at GABAA/B receptors, it may increase whole-brain GABA through unidentified mechanism [44, 45].”

  • The first line medication for MDD are antidepressant drugs. It is interesting to mention allosteric GABA receptor regulators and their potential contribution to MDD management, but the limits and frequent contraindication of GABAergic manipulation in depression should be clearly defined. Ketamine again, displays fast a antidepressant effect which depend on specific homeostatic regulation of the excitatory synapse, as perfectly described in the works of Lisa Monteggia. The link between ketamine, astrocytes and GABA is weak, also taking into consideration that this referee does not believe its activity is contributed by inhibition of NMDA receptors at the level of inhibitory neurons.

Response: We revised as follows.

[lines 472-489] “Additionally, concerns about the risk of dependence, moderate abuse potential, and the danger of withdrawal symptoms, particularly with benzodiazepines, underscore the need for careful consideration of their use [46]. Benzodiazepines bind to the GABAAR at allosteric sites and enhance the GABAAR current by augmenting chloride conductance. Benzodiazepines binding to the synaptic GABAAR induces a conformational change for which GABA has a higher affinity, thereby increasing the frequency of chloride channel opening [47-49]. Benzodiazepines are sometimes used to treat specific symptoms associated with depression, such as anxiety and insomnia. However, a meta-analysis study revealed that treatment with the benzodiazepine alprazolam led to a higher percentage of individuals with MDD achieving response on the 17-item Hamilton Depression Rating Scale (HAM-D) or Clinical Global Impression-Improvement (CGI-I) scale, compared with placebo. Other benzodiazepines, such as diazepam, did not exhibit a clear antidepressant response [50].

In addition, prolonged use of benzodiazepines beyond 2-4 weeks is not recommended due to a potential decrease in GABAergic and monoaminergic function, interference with neurogenesis, and cognitive and psychomotor impairment [51, 52]. These concerns, along with the elevated risk of dependence and suicide attempts, might further restrict the potential use of benzodiazepines in the treatment of depression [53].”

[lines 508-522] “While ketamine's rapid antidepressant effect is primarily attributed to increased glutamate connectivity and signaling, evidence suggests that it induces a transient increase in GABA signaling shortly after administration in rodents [54]. Imaging studies report that ketamine normalizes the hyperactivity of the default mood network (DMN) and altered connectivity with the insula in depressed patients up to 2 days post-infusion, potentially due to enhanced GABA function [55]. Studies in healthy controls support the hypothesis that ketamine enhances GABA inhibition, evidenced by reduced DMN connectivity and decreased reactivity of amygdala-hippocampal circuitry in response to emotional stimuli in ketamine-treated patients [55-57]. Moreover, ketamine increases the GABA/water ratio in the medial prefrontal cortex during the time of ketamine infusion in depressed patients [58]. Both ketamine and its enantiomer, (R)-ketamine, have exhibited swift-acting antidepressant properties and were the subject of active investigation for their potential application in the treatment of MDD [59-61].”

7)  Throughout the paper english mistakes and inappropriate sentences are abundant. For instance: line 72-73 unclear sentence.

Line 248 title of section 4. Line 269 this is absolutely not the consensus. Line 276 “congenital helpless behavior”, what is it? Line 278 “Levels of GABA levels”. Line 422, to be rewritten.

Response: Thank you for the reviewer’s comments. We revised the lines mentioned above as follows:

  • Line 72-73; “GABAA receptors (GABAAR) and GABAB receptors (GABABR). GABAAR is an ionotropic receptor and ligand-gated ion channel responsible for inhibitory synaptic transmission in the CNS [62-64].”

  • Line 248; “Astrocytes and GABA in major depressive disorder (MDD)”

  • Line 269; “Given the prevailing hypothesis that the hypo-GABAergic system is implicated in the molecular mechanisms of MDD, both chemical (SSRI) and non-invasive stimulation such as transcranial magnetic stimulation (TMS) normalize the GABA levels [65, 66].

  •  
  • Line 276; “Zink and colleagues discovered that rats with congenital helpless behavior (cH), a genetic rat model for human depression, exhibited a significant decrease in GAT3 expression compared to non-helpless littermates [67].
  •  
  • Line 278; “Such downregulation might be attributed to the decreased GABA levels in mice with depression-like behaviors.”
  •  
  • Line 422; We re-wrote the whole part of ketamine.

References for response

  1. Fogaça, M.V. and R.S. Duman, Cortical GABAergic Dysfunction in Stress and Depression: New Insights for Therapeutic Interventions. Front Cell Neurosci, 2019. 13: p. 87.
  2. Roberto, M., D. Kirson, and S. Khom, The Role of the Central Amygdala in Alcohol Dependence. Cold Spring Harb Perspect Med, 2021. 11(2).
  3. Liljequist, S. and J. Engel, Effects of GABAergic agonists and antagonists on various ethanol-induced behavioral changes. Psychopharmacology, 1982. 78: p. 71-75.
  4. Vengeliene, V., et al., Neuropharmacology of alcohol addiction. British journal of pharmacology, 2008. 154(2): p. 299-315.
  5. Cousins, M.S., D.C. Roberts, and H. de Wit, GABA(B) receptor agonists for the treatment of drug addiction: a review of recent findings. Drug Alcohol Depend, 2002. 65(3): p. 209-20.
  6. Janak, P.H. and T. Michael Gill, Comparison of the effects of allopregnanolone with direct GABAergic agonists on ethanol self-administration with and without concurrently available sucrose. Alcohol, 2003. 30(1): p. 1-7.
  7. Grant, K.A., et al., Characterization of the discriminative stimulus effects of GABA(A) receptor ligands in Macaca fascicularis monkeys under different ethanol training conditions. Psychopharmacology (Berl), 2000. 152(2): p. 181-8.
  8. Rassnick, S., et al., GABA antagonist and benzodiazepine partial inverse agonist reduce motivated responding for ethanol. Alcohol Clin Exp Res, 1993. 17(1): p. 124-30.
  9. Koob, G.F., A role for GABA mechanisms in the motivational effects of alcohol. Biochemical Pharmacology, 2004. 68(8): p. 1515-1525.
  10. Koob, G.F., A role for GABA in alcohol dependence1. Advances in pharmacology, 2006. 54: p. 205-229.
  11. Koh, W., et al., GABA tone regulation and its cognitive functions in the brain. Nature Reviews Neuroscience, 2023. 24(9): p. 523-539.
  12. Adermark, L. and M.S. Bowers, Disentangling the role of astrocytes in alcohol use disorder. Alcoholism: Clinical and Experimental Research, 2016. 40(9): p. 1802-1816.
  13. Wang, Z., et al., Knowledge atlas of the involvement of glutamate and GABA in alcohol use disorder: A bibliometric and scientometric analysis. Front Psychiatry, 2022. 13: p. 965142.
  14. Liu, J.H., et al., Astrocytic GABA(B) Receptors in Mouse Hippocampus Control Responses to Behavioral Challenges through Astrocytic BDNF. Neurosci Bull, 2020. 36(7): p. 705-718.
  15. Srivastava, I., et al., Blocking Astrocytic GABA Restores Synaptic Plasticity in Prefrontal Cortex of Rat Model of Depression. Cells, 2020. 9(7).
  16. Liu, J., et al., Astrocytes: GABAceptive and GABAergic cells in the brain. Frontiers in Cellular Neuroscience, 2022. 16: p. 892497.
  17. Murphy-Royal, C., S. Ching, and T. Papouin, A conceptual framework for astrocyte function. Nat Neurosci, 2023. 26(11): p. 1848-1856.
  18. Losi, G., L. Mariotti, and G. Carmignoto, GABAergic interneuron to astrocyte signalling: a neglected form of cell communication in the brain. Philos Trans R Soc Lond B Biol Sci, 2014. 369(1654): p. 20130609.
  19. Lee, M., C. Schwab, and P.L. Mcgeer, Astrocytes are GABAergic cells that modulate microglial activity. Glia, 2011. 59(1): p. 152-165.
  20. Kilb, W. and S. Kirischuk, GABA release from astrocytes in health and disease. International Journal of Molecular Sciences, 2022. 23(24): p. 15859.
  21. Lee, M., E.G. McGeer, and P.L. McGeer, Mechanisms of GABA release from human astrocytes. Glia, 2011. 59(11): p. 1600-1611.
  22. Schousboe, A., L. Hertz, and G. Svenneby, Uptake and metabolism of GABA in astrocytes cultured from dissociated mouse brain hemispheres. Neurochemical Research, 1977. 2: p. 217-229.
  23. Schousboe, A. and H.S. Waagepetersen, Role of astrocytes in homeostasis of glutamate and GABA during physiological and pathophysiological conditions. Advances in Molecular and Cell Biology, 2003. 31: p. 461-474.
  24. Schousboe, A., et al., Regulatory role of astrocytes for neuronal biosynthesis and homeostasis of glutamate and GABA. Progress in brain research, 1992. 94: p. 199-211.
  25. Andersen, J.V., A. Schousboe, and P. Wellendorph, Astrocytes regulate inhibitory neurotransmission through GABA uptake, metabolism, and recycling. Essays Biochem, 2023. 67(1): p. 77-91.
  26. Ishibashi, M., K. Egawa, and A. Fukuda, Diverse Actions of Astrocytes in GABAergic Signaling. Int J Mol Sci, 2019. 20(12).
  27. Roberts, E. and S. Frankel, γ-Aminobutyric acid in brain: its formation from glutamic acid. Journal of Biological Chemistry, 1950. 187: p. 55-63.
  28. Tao, R., et al., GAD1 alternative transcripts and DNA methylation in human prefrontal cortex and hippocampus in brain development, schizophrenia. Molecular psychiatry, 2018. 23(6): p. 1496-1505.
  29. Yoon, B.E., et al., Glial GABA, synthesized by monoamine oxidase B, mediates tonic inhibition. The Journal of physiology, 2014. 592(22): p. 4951-4968.
  30. Minelli, A., et al., GAT-3, a high-affinity GABA plasma membrane transporter, is localized to astrocytic processes, and it is not confined to the vicinity of GABAergic synapses in the cerebral cortex. Journal of Neuroscience, 1996. 16(19): p. 6255-6264.
  31. Mederos, S. and G. Perea, GABAergic-astrocyte signaling: A refinement of inhibitory brain networks. Glia, 2019. 67(10): p. 1842-1851.
  32. Mederos, S., et al., GABAergic signaling to astrocytes in the prefrontal cortex sustains goal-directed behaviors. Nature Neuroscience, 2021. 24(1): p. 82-92.
  33. Kang, S. and D.-S. Choi, Astrocyte adenosine signaling and neural mechanisms of goal-directed and habitual reward-seeking behaviors. Neuropsychopharmacology, 2021. 46(1): p. 227.
  34. Hong, S.I., et al., Astrocyte-neuron interaction in the dorsal striatum-pallidal circuits and alcohol-seeking behaviors. Neuropharmacology, 2021. 198: p. 108759.
  35. Thompson, S.M., et al., An excitatory synapse hypothesis of depression. Trends Neurosci, 2015. 38(5): p. 279-94.
  36. Dean, J. and M. Keshavan, The neurobiology of depression: An integrated view. Asian J Psychiatr, 2017. 27: p. 101-111.
  37. Kaltenboeck, A. and C. Harmer, The neuroscience of depressive disorders: A brief review of the past and some considerations about the future. Brain Neurosci Adv, 2018. 2: p. 2398212818799269.
  38. Marx, W., et al., Major depressive disorder. Nat Rev Dis Primers, 2023. 9(1): p. 44.
  39. Guglielmo, R., et al., Editorial: The glutamate hypothesis of mood disorders: Neuroplasticity processes, clinical features, treatment perspectives. Front Psychiatry, 2022. 13: p. 1054887.
  40. Duman, R.S., G. Sanacora, and J.H. Krystal, Altered Connectivity in Depression: GABA and Glutamate Neurotransmitter Deficits and Reversal by Novel Treatments. Neuron, 2019. 102(1): p. 75-90.
  41. Luscher, B., Q. Shen, and N. Sahir, The GABAergic deficit hypothesis of major depressive disorder. Mol Psychiatry, 2011. 16(4): p. 383-406.
  42. Bates, R.E., et al., Retrospective Analysis of Gabapentin for Alcohol Withdrawal in the Hospital Setting: The Mayo Clinic Experience. Mayo Clinic Proceedings: Innovations, Quality & Outcomes, 2020. 4(5): p. 542-549.
  43. Mason, B.J., S. Quello, and F. Shadan, Gabapentin for the treatment of alcohol use disorder. Expert Opinion on Investigational Drugs, 2018. 27(1): p. 113-124.
  44. Morley, K.C., et al., New approved and emerging pharmacological approaches to alcohol use disorder: a review of clinical studies. Expert Opinion on Pharmacotherapy, 2021. 22(10): p. 1291-1303.
  45. Kranzler, H.R., et al., A metaanalysis of the efficacy of gabapentin for treating alcohol use disorder. Addiction, 2019. 114(9): p. 1547-1555.
  46. Manthey, L., et al., Correlates of benzodiazepine dependence in the N etherlands S tudy of D epression and A nxiety. Addiction, 2012. 107(12): p. 2173-2182.
  47. Twyman, R.E., C.J. Rogers, and R.L. Macdonald, Differential regulation of γaminobutyric acid receptor channels by diazepam and phenobarbital. Annals of Neurology: Official Journal of the American Neurological Association and the Child Neurology Society, 1989. 25(3): p. 213-220.
  48. Bianchi, M.T., et al., Benzodiazepine modulation of GABAA receptor opening frequency depends on activation context: A patch clamp and simulation study. Epilepsy research, 2009. 85(2-3): p. 212-220.
  49. Hanson, S.M. and C. Czajkowski, Structural mechanisms underlying benzodiazepine modulation of the GABAA receptor. Journal of Neuroscience, 2008. 28(13): p. 3490-3499.
  50. Petty, F., et al., Benzodiazepines as antidepressants: does GABA play a role in depression? Biological psychiatry, 1995. 38(9): p. 578-591.
  51. Lim, B., et al., Understanding the effects of chronic benzodiazepine use in depression: a focus on neuropharmacology. International Clinical Psychopharmacology, 2020. 35(5): p. 243-253.
  52. Lader, M., Benzodiazepines revisited—will we ever learn? Addiction, 2011. 106(12): p. 2086-2109.
  53. Cutler, A.J., G.W. Mattingly, and V. Maletic, Understanding the mechanism of action and clinical effects of neuroactive steroids and GABAergic compounds in major depressive disorder. Translational Psychiatry, 2023. 13(1): p. 228.
  54. Chowdhury, G.M., et al., Transiently increased glutamate cycling in rat PFC is associated with rapid onset of antidepressant-like effects. Molecular psychiatry, 2017. 22(1): p. 120-126.
  55. Evans, J.W., et al., Default mode connectivity in major depressive disorder measured up to 10 days after ketamine administration. Biological psychiatry, 2018. 84(8): p. 582-590.
  56. Scheidegger, M., et al., Ketamine administration reduces amygdalohippocampal reactivity to emotional stimulation. Human brain mapping, 2016. 37(5): p. 1941-1952.
  57. Scheidegger, M., et al., Ketamine decreases resting state functional network connectivity in healthy subjects: implications for antidepressant drug action. 2012.
  58. Milak, M.S., et al., A pilot in vivo proton magnetic resonance spectroscopy study of amino acid neurotransmitter response to ketamine treatment of major depressive disorder. Molecular Psychiatry, 2016. 21(3): p. 320-327.
  59. Zarate, C.A., et al., A Randomized Trial of an N-methyl-D-aspartate Antagonist in Treatment-Resistant Major Depression. Archives of General Psychiatry, 2006. 63(8): p. 856.
  60. Zhang, K. and K. Hashimoto, An update on ketamine and its two enantiomers as rapid-acting antidepressants. Expert review of neurotherapeutics, 2019. 19(1): p. 83-92.
  61. Hess, E.M., et al., Mechanisms of ketamine and its metabolites as antidepressants. Biochemical pharmacology, 2022. 197: p. 114892.
  62. Jazvinscak Jembrek, M. and J. Vlainic, GABA receptors: pharmacological potential and pitfalls. Current pharmaceutical design, 2015. 21(34): p. 4943-4959.
  63. Sigel, E. and M.E. Steinmann, Structure, function, and modulation of GABAA receptors. Journal of Biological Chemistry, 2012. 287(48): p. 40224-40231.
  64. Watanabe, M., et al., GABA and GABA receptors in the central nervous system and other organs. International review of cytology, 2002. 213: p. 1-47.
  65. Sanacora, G., et al., Increased occipital cortex GABA concentrations in depressed patients after therapy with selective serotonin reuptake inhibitors. Am J Psychiatry, 2002. 159(4): p. 663-5.
  66. Dubin, M.J., et al., Elevated prefrontal cortex GABA in patients with major depressive disorder after TMS treatment measured with proton magnetic resonance spectroscopy. J Psychiatry Neurosci, 2016. 41(3): p. E37-45.
  67. Zink, M., et al., Reduced Expression of GABA Transporter GAT3 in Helpless Rats, an Animal Model of Depression. Neurochemical Research, 2009. 34(9): p. 1584-1593.